# Solid and Liquid Surface-Supported Bacterial Membrane Mimetics as a Platform for the Functional and Structural Studies of Antimicrobials

**DOI:** 10.3390/membranes12100906

**Published:** 2022-09-20

**Authors:** Shiqi Li, Ruohua Ren, Letian Lyu, Jiangning Song, Yajun Wang, Tsung-Wu Lin, Anton Le Brun, Hsien-Yi Hsu, Hsin-Hui Shen

**Affiliations:** 1Department of Materials Science and Engineering, Faculty of Engineering, Monash University, Clayton, VIC 3800, Australia; 2Department of Biochemistry and Molecular Biology, Biomedicine Discovery Institute, Monash University, Clayton, VIC 3800, Australia; 3College of Chemistry & Materials Engineering, Wenzhou University, Wenzhou 325035, China; 4Department of Chemistry, Tunghai University, No. 1727, Sec. 4, Taiwan Boulevard, Xitun District, Taichung 40704, Taiwan; 5Australian Centre for Neutron Scattering, Australian Nuclear Science and Technology Organisation, Locked Bag 2001, Kirrawee DC, NSW 2232, Australia; 6Department of Materials Science and Engineering, School of Energy and Environment, City University of Hong Kong, Kowloon Tong, Hong Kong, China

**Keywords:** monolayer, bilayer, bacterial membrane, quartz crystal microbalance, neutron reflectometry, surface plasmon resonance

## Abstract

Increasing antibiotic resistance has provoked the urgent need to investigate the interactions of antimicrobials with bacterial membranes. The reasons for emerging antibiotic resistance and innovations in novel therapeutic approaches are highly relevant to the mechanistic interactions between antibiotics and membranes. Due to the dynamic nature, complex compositions, and small sizes of native bacterial membranes, bacterial membrane mimetics have been developed to allow for the in vitro examination of structures, properties, dynamics, and interactions. In this review, three types of model membranes are discussed: monolayers, supported lipid bilayers, and supported asymmetric bilayers; this review highlights their advantages and constraints. From monolayers to asymmetric bilayers, biomimetic bacterial membranes replicate various properties of real bacterial membranes. The typical synthetic methods for fabricating each model membrane are introduced. Depending on the properties of lipids and their biological relevance, various lipid compositions have been used to mimic bacterial membranes. For example, mixtures of phosphatidylethanolamines (PE), phosphatidylglycerols (PG), and cardiolipins (CL) at various molar ratios have been used, approaching actual lipid compositions of Gram-positive bacterial membranes and inner membranes of Gram-negative bacteria. Asymmetric lipid bilayers can be fabricated on solid supports to emulate Gram-negative bacterial outer membranes. To probe the properties of the model bacterial membranes and interactions with antimicrobials, three common characterization techniques, including quartz crystal microbalance with dissipation (QCM-D), surface plasmon resonance (SPR), and neutron reflectometry (NR) are detailed in this review article. Finally, we provide examples showing that the combination of bacterial membrane models and characterization techniques is capable of providing crucial information in the design of new antimicrobials that combat bacterial resistance.

## 1. Introduction

Bacteria have been extensively investigated for many decades with respect to the formation of surface colonization and hazards of infection [1]. An untreated bacterial infection can trigger sepsis, leading to tissue damage, organ failure, and death. However, the emergence of antibiotic resistance is a global health challenge, and it is estimated that 10 million people will die annually by 2050 due to the continuous rise in the prevalence of antibiotic-resistant bacteria [2]. Even worse, the last two decades has seen a serious lack of new antibiotics under development due to scientific, regulatory, and financial barriers. In 2019, the World Health Organization (WHO) only classified six innovative antibiotics in clinical development that address the WHO’s list of priority pathogens [2]. The crucial need to develop a new class of antibiotic agents evokes the interest of understanding the interaction of antimicrobials with bacterial membranes.

Bacteria are categorized into the Gram-positive and Gram-negative categories, where they differ in their structures and in the lipid compositions of their cell walls (Figure 1). The major difference between the two categories is the absence of the outer membrane in the Gram-positive cell envelope. Gram-positive bacteria are characterized by the thick cell walls surrounding the cytoplasmic membrane that are composed of multilayers of peptidoglycan (Figure 1a). Through the attachment of the peptide stems to the disaccharide repeat, linear glycan strands are crosslinked into a mesh-like framework [3]. This framework provides protection from the environment and maintains the cell shape and integrity. Gram-positive bacterial membranes are composed of phospholipids, the lipid anchor component (lipoteichoic acid), and various transmembrane and lipoproteins. The typical phospholipids of the plasma membrane include phosphatidylglycerol (PG), cardiolipin (CL), and phosphatidylethanolamine (PE). Depending on the species, the compositions vary substantially. For example, the phospholipids of *Staphylococcus aureus* predominantly contain PG with a large amount of lysyl-phosphatidylglycerol (L-PG) [4]. In *Bacillus subtili*, PG and PE are major components without the existence of L-PG [4]. 

The Gram-negative bacterial cell envelope consists of the inner membrane, periplasm, and outer membrane (Figure 1b). The inner membrane comprises phospholipids, predominantly PE, PG, and CL. A thin layer of peptidoglycan is located in the periplasm, which supports the shape and rigidity of the bacterium. The outer membrane is extremely asymmetric, with a variety of proteins embedded in the membrane that serve as an additional selectively permeable barrier for Gram-negative bacteria [5,6]. *Escherichia coli* (*E. coli*) is the most well-known of the Gram-negative bacteria species that have been extensively studied [7,8]. Thus, various recent studies have attempted to fabricate Gram-negative bacterial membrane models based on the compositions and structures of *E. coli*, which are amenable to physical and biophysical investigations. Phospholipids are the main component of the inner leaflet of the outer membrane, and the outer leaflet predominantly consists of lipopolysaccharides (LPSs) [5,9]. In LPSs, lipid A is covalently attached to the core polysaccharide region. Lipid A comprises six acyl chains linked to a phosphorylated N-acetylglucosamine (ClcN) headgroup [6]. For diverse bacterial species, the composition of the core polysaccharide region is consistent [5]. The core region is connected to an O-antigen, which is a long chain of LPS with high variability among bacterial strains. Most natural Gram-negative bacteria express smooth LPSs, containing the O-antigen. There are also rough mutant strains, which are engineered to consist of lipid A and either complete or only truncate the core region [8].

Bacterial membranes have various compositions and hence possess diverse physiochemical properties. Due to the dynamic nature, complex compositions, and small sizes of real bacterial membranes, the advancement of detailed structural information and in vivo examination of membrane dynamics has been hindered [8,10]. In order to investigate the properties, structures, and transport mechanism across the membrane, a number of model bacterial membranes have been developed with well-defined and controllable compositions. A model bacterial membrane system allows for a biophysical and structural investigation at the molecular scale, and the use of such platforms facilitates the understanding of the interactions of novel antibiotics with the bacterial membrane [11]. Biomimetic bacterial membranes include lipid monolayers [12,13,14], solid-supported lipid bilayers [15,16,17,18], spherical liposomes [19,20,21], and free-standing planar lipid membranes [22,23,24]. In this review, we only focused on lipid monolayers and solid-supported lipid bilayers because of their stability in air, and the ease of forming and manipulating their lipid compositions [10,25]. Depending on the compositions of two leaflets, the solid-supported lipid bilayers can be further characterized as symmetric bilayers and asymmetric bilayers with an increasing similarity to the Gram-negative bacterial outer membrane. Additionally, we present the most common methods that have been utilized to fabricate these models.

Antimicrobials which specifically combat infections caused by bacteria are mentioned here, i.e., antibiotics. The interactions between antibiotics and their bacterial-specific target are complex and antibiotics can affect different cellular components [26]. As the focus of this review was on the applications of mimetic bacterial membranes, we discus two surface-active last-line antibiotics, daptomycin [27], and polymyxin B [28]. We also discuss other new emerging alternatives for conventional antibiotic therapies, such as antimicrobial peptides, lipid nanoparticles such as cubosomes, and medium-chain fatty acids and monoglycerides. Solid and liquid surface-supported bacterial membrane mimetics can facilitate the investigation into how they interfere with bacterial membranes. To probe the membrane formation process, membrane structures, and their interactions with biological molecules, a variety of surface-sensitive investigation techniques have emerged, and we examined three techniques in this review: quartz crystal microbalance (QCM), surface plasmon resonance (SPR), and neutron reflectometry (NR). We also summarize examples of the use of model bacterial membranes to study the structures, dynamic properties, and biomolecular interactions.

## 2. Biomimetic Bacterial Membrane Systems

### 2.1. Lipid Monolayer to Mimetic Bacterial Membrane

The lipid monolayer model is one of the most common biomimetic systems used to study the protein–lipid and drug–lipid interactions. It is a simplified model to mimic half of the leaflets of the natural membrane by using the Langmuir–Blodgett technique. The formation of the Langmuir monolayer on the liquid subphase is attributed to the amphiphilic properties of phospholipids, orienting themselves to minimize their free energy at the gaseous and liquid interface [29].

The unique advantage of the lipid monolayer model is the ability to precisely control the lipid composition at the interface and change the molecular areas of the lipids [30]. Thus, it allows for a rigorous thermodynamic analysis via surface tension measurements [31,32,33]. Surface tension is induced by an unbalanced molecular attraction on surface water molecules toward the bulk subphase, which occurs to minimize contact area with the air [29]. The addition of amphiphiles to the water surface causes change in the surface tension, which allows for the characterizations of the lipid–lipid and lipid–water interactions [34]. Upon the compression of the Langmuir monolayer by movable barriers along the water surface, the surface pressure (π) is continuously measured as a function of the mean molecular area (A) to provide further information about the lipid–lipid, lipid–water, or lipid–drug interactions [30,34,35]. Moreover, the Langmuir monolayers have been broadly utilized because of their homogeneity, well-defined geometry, stable structure, and specific orientation. Apart from the composition and molecular packing, the Langmuir technique permits the manipulation of the physical states, lateral pressure of the membrane, and experimental conditions, such as temperature or pH [31,36].

The main limitation of the Langmuir monolayer is associated with the fact that it only represents one leaflet of a bacterial membrane. Thus, it is not suitable to study transmembrane processes, although this biomimetic monolayer system is certainly valuable to investigate processes occurring at the membrane’s surface. Therefore, the Langmuir monolayer is applicable in the study of intermolecular behavior, the effects and interactions of compounds such as antimicrobial peptides on the membranes, and in the study of how certain parameters impact the interactions, such as cations strength or temperature [11,31,36,37,38]. The simplification of the monolayer membrane system is necessary to study specific interactions at the molecular level. However, it can impede the accurate understanding of some membrane functions.

### 2.2. Lipid Bilayer to Mimetic Bacterial Membrane

#### 2.2.1. Symmetric Bilayer System

The symmetric bilayer model, whose inner and outer leaflets have identical compositions, is widely used to model the structure of bacterial membranes. For a closer mimicry, the mimic bilayer leaflets are constructed with the aim of being identical to the bacterial membrane [39]. Symmetric bilayers have prominent advantages regarding the modeling of bacterial membranes: they are simple to prepare and can be used for a wider variety of characterization techniques [16,18,40,41,42].

Symmetric bilayers are widely applied to model bacterial membranes for the investigation of protein–lipid interactions, such as antibacterial peptide activities. For instance, 1,2-dimyristoyl-sn-glycero-3-phosphocholine/1,2-dimyristoyl-sn-glycero-3-phosphoryl-3′-rac-glycerol (DMPC/DMPG) bilayers are applied to create simple model bacterial membranes that reflect the overall negative charge typically found on bacterial membranes. Such model membranes have been used to study the interactions of the antimicrobial peptide LL-37 loaded into nanoparticles [43]. The application of symmetric bilayers renders reproducing the lipid composition easy and allows for the monitoring of molecule interactions [44]. However, natural cell membranes are composed of asymmetric bilayers [40], which means that the phospholipid compositions of the inner and outer leaflets are different. While a well-synthesized symmetric bilayer can be used to aid in the understanding of bacterial membranes, asymmetric bilayers are capable of providing a closer mimic to the intrinsic natural membranes.

#### 2.2.2. Asymmetric Lipid Bilayer

The outer membrane of Gram-negative bacteria is considered to be an additional barrier against antibiotics and hence biophysical and structural studies of the outer membrane are of significant interest [45]. Asymmetric lipid bilayer models have been created to serve as more realistic models for more advanced studies of the Gram-negative bacterial outer membrane at the molecular level.

The combination of Langmuir–Blodgett and Langmuir–Schaefer techniques is a method that has been reported to successfully form asymmetric bilayer membranes with high coverage [6,8]. Phospholipids, generally phosphatidylcholine (PC), were deposited as an inner leaflet by the Langmuir–Blodgett and Langmuir–Schaefer deposition techniques, and lipid A or *E. coli* crude LPS outer leaflets were deposited on an ultra-flat silicon wafer surface [8,46], allowing for the composition of each bilayer leaflet to be precisely controlled [47,48]. Those asymmetric solid-supported bilayers can be used to analyze the lipid phase separation and lipid domain formation in mixed bilayers [49]. They can also be used for the study of protein and peptide binding to LPSs within a complete bilayer environment [50,51,52].

The solid-supported bilayers have a significant limitation in that there is only a very thin layer of water between the lipid bilayer and the solid support [53]. The insufficient space hampers the incorporation of transmembrane proteins and leads to an inability to have the incorporated membrane proteins freely diffuse through the model membrane [54,55,56]. Thus, tethered bilayer lipid membranes have been developed to provide appropriate space between the bilayer and solid support by using anchorlipids to covalently anchor the lipid bilayer to the solid substrate surface [57,58,59]. The tethered bilayer lipid membranes facilitate the accommodation of membrane proteins into the bilayer, but the packing density of the proximal leaflet can have adverse effects on the function of the membrane proteins. The tethered bilayer membranes can be enhanced by mixing anchorlipids with spacer molecules, diluting the tethering density [57,58]. This model can be exploited to study the interaction of antibiotics with the bacterial membrane, such as colistin [60].

## 3. Formation of Various Bacterial Model Membranes

### 3.1. Langmuir–Blodgett Technique to Form Lipid Monolayer

In the last few decades, the Langmuir technique has been successfully utilized to produce model bacterial monolayer surfaces, and the process of Langmuir monolayer formation is illustrated in Figure 2. The monolayers are formed by using Langmuir troughs which are usually made of Teflon and have one or two movable barriers. The Langmuir trough also contains one pressure sensor to measure the surface pressure (π), mostly using a Wilhelmy plate [29]. To form a monolayer, a small volume of a diluted solution of amphiphilic compounds is deposited in a volatile and water-immiscible solvent on a water or aqueous buffer surface. Subsequently, the diluted solution will spread over the available area of the interface. After evaporating the organic solvent, the hydrophilic (water soluble) headgroups of the phospholipids make contact with the water and the hydrophobic (water insoluble) tails remain in the gas phase due to the amphiphilic properties of phospholipids. This results in the formation of an initial one-molecule-thick lipid film with a low packing density (Figure 2a). The film-forming ability depends on the balance between the size of the hydrophobic portion and the strength of the hydrophilic headgroups. At the start, the amphiphilic molecules are far enough apart that the hydrophobic tails are distributed near the interface and the monolayer exerts little effect on reducing the surface tension of the aqueous subphase. If the mobile barriers compress the molecules at the interface at a constant rate to reduce the available area to the monolayer, the surface pressure increases (Figure 2b). During the compression, the hydrophobic tails begin to interact, which increases the surface pressure; the amphiphiles undergo self-organization until the individual molecules are perfectly oriented in a vertical position (Figure 2c). The pressure sensor continuously monitors the surface pressure (π) and the data are used to conduct a π-A isotherm, which is a plot of the change in π as a function of the average area per molecule at the interface (Figure 2d). The isotherm indicates the monolayer formation, phase transitions, and conformational transformations during the compression process, and also details the monolayer’s properties.

A range of Langmuir-type monolayers have been successfully fabricated with different compositions [61,62,63,64,65]. Langmuir monolayers composed of pure bacterial phospholipids (PE, PG, CL) or mixtures of phospholipids have been used as model membranes for Gram-positive membranes and Gram-negative inner membranes [61,66]. Models of Gram-negative bacterial outer membranes have used pure lipid A [64,67], deep rough LPS mutant (Re-LPS) [62,64], or smooth LPS [63,65]. Le Brun et al. used the Langmuir monolayer to mimic a Gram-negative bacterial outer membrane where a pure rough *E. coli* Rc-LPS monolayer and placed it at the air–liquid interface to emulate the cell surface. This was formed by depositing the solution of Rc-LPS dropwise onto the liquid surface of the Langmuir trough [5]. A pressure–area isotherm was then conducted via repeated compressions and relaxations of the movable Langmuir barriers. The isotherm demonstrated that the Rc-LPS monolayer was able to achieve high surface pressures of over 40 mN m^−1^, beyond the possible pressure range of a natural Gram-negative bacterial membrane [5]. Additionally, the stability of the monolayer was proven by a small change in surface pressure at a fixed barrier area. Therefore, Langmuir monolayers are capable of being used as model bacterial membrane surfaces with realistic surface topology and molecular fluidity [5]. These Langmuir monolayer models have been extensively used to explore the interaction of antimicrobial agents with the bacterial membrane. For example, a monolayer of Ra-LPS from a *Salmonella enterica* rough mutant, resembling a Gram-negative bacterial outer membrane, was spread over the interface of the Langmuir film balance to understand the influence of aromatic alcohol and cationic surfactants on bacterial outer membrane structures [68].

### 3.2. Vesicle Fusion Method to Form Supported Lipid Bilayers

Vesicle fusion is a preferred method to form solid-supported lipid bilayers (SLBs) owing to the simplicity of the method [69]. The formation of small unilamellar vesicles by tip sonification or extrusion through a filter is already well understood [15,69,70]. As shown in Figure 3, the vesicle fusion method for forming a solid-supported bilayer is composed of three stages, namely vesicle adsorption, fusion, and bilayer formation [70,71,72]. At the initial stage, small unilamellar vesicles are adsorbed to the substrate surface, such as silica or mica surfaces (Figure 3a). Next, a spontaneous fusion of small unilamellar vesicles (100–200 nm) occurs, which forms small patches of SLBs on the substrates (Figure 3b) [70]. Finally, continuous SLBs can be formed by the spread of the small patches (Figure 3c) [70]. This method has been used on dextran modified with lipophilic compounds existing on the gold surface of the L1 sensor chip (BIAcore, Cytiva), which can trap liposomes to the sensor surface for the synthesis of SLBs [44,73]. These sensor chips are widely used due to the shortened process time and because the process is easy to repeat. We will provide more details in Section 4.2.

Recent research studies have shown some successful cases in forming SLBs by the vesicle fusion method to mimic bacterial membranes. For example, the SLBs composed of *E. coli* phospholipid extract was assembled on a silica surface, showing that the surface charge, pH, and fusion promoter concentration can also affect the quality of the bilayer formation [69]. It is worth noting that a fusion promoter such as a divalent cation can facilitate the adsorption of negatively-charged small unilamellar vesicles to an anionic substrate surface of mica [74]. In addition, high-quality SLBs can be fabricated by the vesicle fusion method below the phase transition temperature [70]. A recent study demonstrated that it was possible to form SLBs by placing dipalmitoylphosphatidylcholine (DPPC) at the deposition temperature below the melting point of the lipid to mimic cytoplasmic bacterial membranes, which was possible due to the presence of continuous flow, allowing for an efficient transfer of small unilamellar vesicles to substrate surface [70].

### 3.3. Langmuir–Blodgett and Langmuir–Schaefer Approach to Form Asymmetric Bilayers

Tamm and McConnell [53] developed a method to fabricate lipid bilayers on hydrophilic silicon substrates, combining the vertical dipping Langmuir–Blodgett (LB) deposition technique with the horizontal dipping Langmuir–Schaefer (LS) technique (Figure 4). Mimetic asymmetric bilayer membranes are usually composed of phospholipids in the inner leaflet and rough LPS in the outer leaflet regions [46] and can be fabricated using LB/LS dipping. Initially, a self-assembled phospholipid monolayer at the air–water interface is compressed to a suitable surface pressure in a Langmuir trough (Figure 4a) [29,69]. The lipid monolayer is transferred by immersing the hydrophilic silicon wafer into a subphase and then vertically lifting the hydrophilic substrate by the Langmuir–Blodgett deposition method (Figure 4b), creating the lower leaflet of the bilayer [29]. The LPS monolayer can then be spread at the air–water interface and be compressed into a dense monolayer. By using the horizontal Langmuir–Schaefer method, the substrate containing a phospholipid monolayer is lowered until it makes contact with the LPS monolayer (Figure 4c), and then it is pushed across the interface to deposit the second layer (Figure 4d) [29], thus completing the bilayer. 

In addition to the single-supported asymmetric bilayers, the combination of Langmuir–Blodgett and Langmuir–Schaefer deposition techniques can be exploited to form floating membranes or supported double bilayers. The supported double bilayers have a maximum of 20–30 Å inter-membrane water reservoirs when the double bilayers are fabricated with phosphatidylcholine [75]. The floating bilayer is not restrained by the solid substrates and is suitable for transmembrane studies. However, Stidder et al. showed that not all phospholipids can form stable double bilayers [76]. Double bilayers of 1,2-dipalmitoylphosphoethanolamine (DPPE) are stable in the gel phase, but are unstable in the fluid phase as a planar structure [76]. Currently, the supported double bilayers have not been applied in bacterial membrane mimetics studies. Instead, a hybrid floating membrane model has been developed, resembling Gram-negative bacterial membranes. Clifton et al. investigated the asymmetrical outer membrane structure and dynamics of Gram-negative bacteria by using outer membrane model floating membranes [77]. The inner phospholipid leaflet was deposited onto the ω-thiolipid-coated substrates by the Langmuir–Blodgett deposition technique, followed by a deposition of Ra-LPS as the outer leaflet layer via the Langmuir–Schaefer transfer to mimic a Gram-negative outer membrane. This hybrid floating membrane preserves the inter-membrane water layer between the asymmetric bilayer and silicon wafer, which reduces the frictional impacts of the solid support on the membrane [78]. The close corroboration between simulations and the performance of the experimental model systems validates that this floating outer membrane model is a powerful tool to help learn the biological and biophysical behaviors of bacterial membranes [77,78]. However, the outer membrane is connected to the periplasm by proteins and is not freely floating. Thus, the hybrid floating membrane model cannot represent the exact behaviors of real outer membranes.

## 4. Surface Characterization Techniques Used to Determine the Membrane Mimetic Systems

### 4.1. Quartz Crystal Microbalance with Dissipation

Quartz crystal microbalance with dissipation (QCM−D) is an acoustic surface-sensitive technique based on quartz crystal microbalance technology (Figure 5) [43]. QCM−D has been widely applied to assess supported lipid bilayer (SLB) formation by vesicle fusion and probe the biomolecular interactions with the surfaces within the liquid. In the SLB formation process (Figure 5b), upon adsorption of the vesicles or bilayer to the sensor crystal surface, the shifts in resonance frequency (Δf) correlate with the mass change. Through the Sauerbrey equation, the mass of the substance adsorbed on the crystal sensor can be related to the frequency change as follows:(1)$$\Delta f=-\frac{2f_{0}{}^{2}}{A\sqrt{\rho_{q}\mu_{q}}}\Delta m$$
where Δf is the frequency shift, Δm is the mass change, $f_{0}$ is the initial resonance frequency of the crystal, A is the active area of the crystal, and $\rho_{q}$ and $\mu_{q}$ are the density and shear modulus of quartz, respectively [79]. Measuring the change in the resonant frequency of the crystal will therefore provide information on the mass change and surface coverage, making it well suited for the characterization and analysis of biomimetic membranes [80,81]. The changes in energy dissipation (Δd) provide the real-time properties of the softness or viscoelasticity of the films deposited on the chip surface, which makes it possible to monitor the conformational changes. The articles of Hook et al., Alexander et al., and Edvardsson et al. provide more details about the QCM−D technique [82,83,84].

The main application of the QCM−D technique in the characterization of biomimetic lipid layers of bacterial membranes is during the deposition phase, where vesicle layer formation can be monitored [86]. For example, Lind et al. used the QCM−D technique to study the formation of an SLB composed of a native total lipid extract of *E. coli* [69]. The QCM−D responses indicated that the divalent cation, lipid concentration, vesicle injecting flow rate, and chamber temperature profoundly affected the success of forming the SLB [69]. Once SLBs are formed, the QCM−D technique can be further exploited to investigate the binding and interaction between biomolecules, as well as the liquid surface via subtle changes in frequency and dissipation. For example, in Hsia et al.’s study, the QCM−D technique was used to monitor antibiotic interactions between the outer membrane-like supported bilayer and polymyxin B [87]. By elucidating the frequency and dissipation changes with a two-layer mechanical model, the authors were able to capture the disruption kinetics and changes in the membrane mechanical properties after the binding of polymyxin B [87].

### 4.2. Surface Plasmon Resonance

Surface plasmon resonance (SPR) is a real-time, label-free method which has been widely utilized in clinical analysis for biomolecular interactions [88]. The SPR method relies on the surface plasmon resonance phenomenon. When an incident light strikes on the interface of materials with a different refractive index, a portion of light, evanescent light, will pass through the surface of the medium with a lower refractive index instead of being entirely reflected. In SPR, electrons within the metal surface of a metal-dielectric interface are stimulated by the evanescent light, yielding surface plasmons that propagate parallel to the metal surface [89]. The angle at which the resonance occurs is dependent on the refractive index of the materials close to the metal surface. Any alteration in the refractive index of the sensing material can prevent the formation of the plasmon, which allows for the quantification of the amount of analytes by detecting the changes in the light intensity or by detecting the resonance angle shifts of reflected light [90]. In general, a commercially available SPR instrument contains an optical light source, a flow channel, a sensor chip, and a detection system (Figure 6a) [89]. The sensor chip comprises a thin layer of gold coupled to a glass layer and the gold–glass dielectric layer is attached to a prism. The HPA and L1 chips (from BIAcore, Cytiva) have often been used in the membrane formations of SPR applications (Figure 6b). The sample is injected through the flow channel and interacts with the ligands that are immobilized on the sensor’s surface. The incident light from the light source is reflected onto the gold surface and the reflected light is captured by the diode array detector [29]. The change in signal is illustrated in the form of a sensorgram (Figure 6c). The change in optical properties is dependent on the thickness of the gold layer, the light wavelength, and the adsorbed mass on the sensor surface [88].

The core of an SPR instrument is its sensor chip, which allows for the interactions between the analytes and immobilized membrane system. There are two commercially available chips from BIAcore International, the HPA and L1 chips (Figure 6b) which are used for phospholipid membrane formation [44]. To generate HPA chips, a flat hydrophobic alkanethiol surface is prepared by covalently attaching hydrophobic alkanethiol molecules to the thin gold layer. Next, liposomes are applied to the hydrophobic surface and then liposomes are trapped onto the surface to form a lipid monolayer. However, in the HPA chip, the insertion of biomolecules into the hybrid lipid membrane can be hindered by the covalent attachment of the monolayer with the gold surface [44]. Hence, the L1 chip was made to resemble the fluid bilayer structure and enable the analysis of the penetration of biomolecules into the membrane. A thin dextran layer with exposed lipophilic compounds is coated onto the L1 chip. The applied liposomes are adsorbed to the surface using the lipophilic compounds, forming a lipid bilayer system [44]. The immobilization of the mimetic lipid membrane on the gold layer enables both sensor chips to study the biomolecular interactions with the membrane. More information regarding the SPR technique can be found in the reviews of Nguyen et al. and Hall et al. [44,88].

SPR technology can provide extensive information on the binding affinity, binding specificity, dissociation and association rate constants, and key thermodynamic parameters of molecular binding mechanisms [88]. Therefore, the SPR method is suitable for the investigation of membrane-binding properties of antimicrobial peptides. For instance, an SPR-based biosensor was employed to study the interactions of daptomycin with four biomimetic lipid membranes of a mammal cell along with Gram-positive and Gram-negative bacteria. The sensorgrams reveal that daptomycin has a higher binding affinity toward the Gram-positive bacterial membrane of 1-palmitoyl-2-oleoylphosphatidyl-glycerol (POPG) and 1-palmitoyl-2-oleoylphosphatidyl-glycerol/ cardiolipin (POPG/CL) at a volumetric ratio of 1:1 on the L1 chip. Meanwhile, the binding kinetics of Gram-positive bacteria was evaluated by the SPR method [92]. The affinity data and sensorgrams reveal that the Gram-positive bacterial membrane of POPG more slowly dissociate daptomycin than the POPG/CL membrane, and daptomycin dissociates quickly from both membranes in the absence of calcium ions. This established SPR method offers the possibility to estimate the binding characteristics of antimicrobial agents to model bacterial membranes [92]. The SPR method is also utilized in many studies to assess the binding properties of linear cationic amphipathic peptides, cyclic antimicrobial peptides, and glycopeptide antibiotics with the bacterial membranes [44].

### 4.3. Neutron Reflectometry

Neutron reflectometry (NR) is a technique used for the measurement of the structure and composition of lipid membranes. Compared with other available techniques, NR has advantages in its ability to determine the complex lipid compositions of membranes with high accuracy and no damage [39]. In addition, the NR technique is able to identify the relative location of lipids and proteins in the membranes with multiple components by the deuteration and contrast matching techniques [39]. The use of hydrogenated and deuterated molecules allows for the highlighting of individual components due to the difference in contrast between the lipids/proteins of interest and the bulk isotopic composition of the solvent [46,93]. The NR technique can be used as a complementary technique to QCM and SPR as it provides detailed information concerning the penetration of biological molecules across the membrane and does not only detect information on the surface of the membrane. For this technique, a neutron beam is directed at the samples and then reflected (Figure 7) [94]. There are two types of reflected signals, specular reflection and off-specular reflection. The incidence angle and reflection angle of specular reflection are identical, which can be used to measure the thickness of the membrane and structure perpendicular to the plane of the surface [39]. An unsmooth surface possibly generates an off-specular reflection, which has different incidence angles and reflection angles. An in-plane structure such as a lipid domain formation could be determined using off-specular reflection, however the signal is usually very weak [39], and hence most studies are based on specular reflection [94]. For specular reflection, the reflectivity (=reflected intensity/incident intensity) is measured as a function of the magnitude of the scattering vector, *q_z_*, which is defined in Equation (2) [94]:(2)$$q_{z}=\frac{4\pi sin\theta }{\lambda }$$
where θ refers to the incidence angle and λ refers to the neutron wavelength. In order to perform the analysis, the bacterial membrane models are described as a series of layers of homogeneous material; typically, the layers that are used are inner leaflet headgroups, inner leaflet tails, outer leaflet tails, and outer leaflet headgroups. Each layer possesses a distinct thickness, scattering length density (SLD), and roughness. The SLD modulates the refractive index of a material for neutrons and is dependent on the composition of the model membrane.

The NR technique can be conducted at the air–water interface or the solid–liquid interface. This means that both lipid monolayers and solid-supported lipid bilayers can be measured. However, to ensure the best results, there are some constraints on the sample quality [95]. For instance, solid substrates in the supported membrane must be polished until they are smooth (typically rms < 5 Å) due to interfacial roughness being the dominant factor that limits the quality of the data [39]. In addition, ultra-thin solvent reservoirs can be applied to reduce the incoherent background to increase the measurable *q* range [96,97]. More details concerning the principles and applications of NR can be found in Cousin and Fadda’s journal article [98].

## 5. Applications: Gram-Positive Bacterial Membranes

### 5.1. Different Mimetic Gram-Positive Bacterial Membranes

Examples of Gram-positive bacteria include *Staphylococcus aureus*, *Streptococcus pneumoniae*, *Enterococcus faecalis*, and *Bacillus subtilis*. The compositions of the cellular membrane of these species are diverse, but they contain major components of PG and CL [99]. At least one type of aminoacylatedphosphatidylglycerol can be found in many Gram-positive species [99]. The composition of the head groups and the fatty acyl chains in phospholipids can substantially change to adapt to a particular environment [100]. Therefore, it is essential to construct an accurate lipid composition to study the properties, dynamics, and interactions of antimicrobials with the Gram-positive bacterial membrane. 

A variety of lipid mixtures have been used to mimic bacterial membranes, including different ratios of phospholipids. The choice of lipids depends on the properties of the lipids, ease of lipid bilayer formation, nature of the lipid phase, and biological relevance [101]. PC/PG, PE/PG, and CL/PG model membranes at 1:1(mol/mol) ratios have been considered as possible models for Gram-positive bacteria such as *S. aureus* and *Bacullus cereus* [101]. In Kinouchi’s study, pure PG and PG/CL at a ratio of 1:1 (*v*/*v*) were used to prepare small unilamellar vesicles which were then deposited onto a L1 sensor chip surface to form a mimetic Gram-positive bacterial membrane for SPR studies [92]. Jiang et al. used PG, CL, and L-PG at the molar ratios of 69:12:19 and 23:60:17 to construct symmetric membrane bilayers using the Langmuir–Blodgett and Langmuir–Schaefer approaches corresponding to the *S. aureus* wild-type and mutation strains, respectively [102]. The lipid ratios were obtained from the lipidomics results that were the most accurate lipid compositions for the Gram-positive bacterial mimetic systems.

### 5.2. Interaction of Antimicrobials with Gram-Positive Bacterial Membranes

The increasing number of antibiotic-resistant strains of bacteria poses challenges for treating infections worldwide. The Gram-positive pathogen, *S. aureus*, has a considerable capability to develop antibiotic resistance and the increase of antibiotic resistance triggers a significant reliance on last-line antibiotics, such as daptomycin [102]. Daptomycin is the only clinically approved membrane-active antibiotic that can combat infections caused by Gram-positive bacteria [103]. Membrane-active antibiotics are proposed to have slower resistance development and have evoked increased and renewed interest [103]. Therefore, the studies of the mechanism and interactions of daptomycin with model Gram-positive bacterial membranes facilitate novel drug development. Mescola et al. investigated the effects of daptomycin on the structure of the supported lipid bilayers of a 1-palmitoyl-2-oleoyl-*sn*-glycero-3-phosphoethanolamine (POPE)/ 1-palmitoyl-2-oleoyl-*sn*-glycero-3-phospho-(1′-*rac*-glycerol) (POPG) lipid mixture at a molar ratio of 3:1 and a 1-palmitoyl-2-oleoyl-*sn*-glycero-3-phosphocholine (POPC)/ 1,2-dimyristoyl-*sn*-glycero-3-phospho-(1′-*rac*-glycerol) (DMPG) lipid mixture at a molar ratio of 1:1 on mica substrates [104]. The fluorescence microscopy and atomic force microscopy images of the model membranes exposed to daptomycin demonstrated that daptomycin predominantly interacted with the PG-rich region and penetrated more deeply into the bilayer in this region, triggering a change in the phase state [104]. To achieve more realistic lipid compositions of a Gram-positive bacterial membrane, CL was added in some studies. For example, 1-palmitoyl-2-oleoylphosphatidyl-glycerol (POPG) and 1-palmitoyl-2-oleoylphosphatidyl-glycerol/ cardiolipin (POPG/CL) mixtures have been used to mimic a Gram-positive bacterial membrane for the delineation of the membrane-binding properties of daptomycin [92]. The results showed that daptomycin initially bound to the hydrophilic group, then became inserted in the lipid bilayers [92]. In a study investigating the interactions of daptomycin with a model membrane of 1-palmitoyl-2-oleoyl-*sn*-glycero-3-phosphoglycerol/ 1,2-dipalmitoyl-*sn*-glycero-3-phosphoglycerol/ 1′,3′-bis [1,2-dimyristoyl-*sn*-glycero-3-phospho]-glycerol (POPG/DPPG/CL = 1:1:2 molar ratio), the surface pressure of the monolayer was primarily monitored to indicate daptomycin’s ability to bind and insert into the lipid film, change the packing density of lipids, and then induce a moderate fluidization of the monolayer [105]. To further investigate the interactions, a solid-supported lipid bilayer was formed in a QCM-D chamber. The QCM-D data showed that daptomycin quickly aggregated on top of the membrane, followed by a slow insertion into the bilayer; a change in the mass and frequency demonstrated that majority of the daptomycin accumulated in the upper leaflet of the model membrane [105]. Other non-membrane targeting antibiotics, such as aminoglycosides, macrolides, and fluoroquinolones are found to contain weak interactions with bacterial membrane models [106] and thus we will not discuss these antibiotics further in this review.

Resistance to daptomycin is induced by a metabolic adaptation strategy of *S. aureus* strains. The adaptation strategy causes Cls2 point mutations in genes, which reduce PG and increase CL contents in the bacterial membrane to evade daptomycin and innate immune cell attacks [102]. *S. aureus* wild-type (PG:CL:L-PG = 69:12:19 molar ratio) and CL-rich mutation (PG:CL:L-PG = 23:60:17 molar ratio) membranes were reconstituted on a silica surface using the Langmuir–Blodgett and Langmuir–Schaefer methods. Both membranes were further used to examine how membrane compositional/structural change impedes daptomycin bactericidal functions using the neutron reflectometry technique. Upon interaction with daptomycin, the fringe of the neutron reflectivity curves of the wild-type membrane were significantly shifted, indicating the extraction and solubilization of the mimetic *S. aureus* wild-type membrane. In contrast, in the presence of daptomycin at a CL-rich mutation membrane, the fitted neutron data did not find any membrane solubilization [102]. Those findings were well correlated to the bacterial membrane adaptation strategy, providing a potential novel therapeutic approach in the treatment of *S. aureus* resistance.

Apart from novel therapeutic targeting research, in recent years a new class of antimicrobial peptides has shown to be a potent solution due to its ability to physically disrupt the bacterial membrane. The advantage of antimicrobial peptides over current antibiotic drugs is that they are less prone to suffering from drug resistance because of its broad spectrum of activity [107]. Numerous studies have been conducted to investigate the interaction of antimicrobial peptides with the Gram-positive bacterial membrane. For example, the interactions of alamethicin with lipid membranes were explored using the QCM-D technique [108]. Lyophilized powder L-α-phosphatidylcholine (Egg PC) was used to prepare the liposomes. The formation of the supported lipid bilayer on the quartz crystal in an aqueous environment employed the vesicle deposition method. The changes in frequency and dissipation demonstrated a vertical insertion of the peptide. The insertion forms cylindrical cores on membranes and evokes a significant disordering of lipids [108]. The SPR method is also a popular technique used to analyze the antibiotic activity of antimicrobial peptides. The activity of the glycopeptide against Gram-positive bacteria is induced by the ability to bind mucopeptide precursors that terminate in the sequence KAA [109]. PC vesicles were loaded to the HPA chip and doc-KAA was subsequently inserted into the lipid monolayer to form a ligand–lipid monolayer. The binding affinity constants of the glycopeptide were measured by the SPR method, and the binding of glycopeptides to the model membrane was modulated by the dimerization and membrane anchoring of the antimicrobial peptides [109].

Another promising antibacterial candidate to become a next-generation bacterial infections treatment are antimicrobial lipids. Antimicrobial lipids are single chain amphiphiles with the capacity to destabilize bacterial membranes. Similar to antimicrobial peptides, it is difficult for bacteria to develop drug resistance to antimicrobial lipids [110]. Particularly, medium-chain saturated fatty acids (8–12 carbon atoms) and monoglyceride derivatives (esterified acid with an adduct of a glycerol molecule) have received attention because of their great antibacterial potency [111,112]. It was found that lauric acid (C-12) exhibits potent antibacterial activity against the growth of Gram-positive bacteria. This inhibitory activity is magnified for its monoglyceride, glycerol monolaurate [113]. To obtain the molecular-level interaction kinetics of medium-chain saturated fatty acids and monoglycerides, SLBs have been employed in several studies, which allow for the real-time monitoring of the interactions [114,115,116]. Yoon et al. formed a supported lipid bilayer of 1,2-dioleoyl-sn-glycero-3-phosphocholine (DOPC) on the silicon dioxide surface to investigate the effects of lauric acid (anionic) and glycerol monolaurate (nonionic) on the membranes [116]. The QCM-D technique was used to monitor the changes in the mass and viscoelastic properties of the SLB upon treatment. Both lauric acid and glycerol monolaurate induced membrane disruption above the respective critical micelle concentration values [116]. At high concentrations, the QCM-D results of lauric acid showed a simultaneous decrease in frequency shifts and an increase in energy dissipation shifts, and at lower concentrations there were minor changes. By contrast, glycerol monolaurate exhibited rapid changes in frequency shifts and corresponding energy dissipation shifts at high concentration; at lower concentrations, the change was more gradual but still significant. The QCM-D results were consistent with fluorescence microscopy measurements and minimum inhibitory concentration tests against *S. aureus*. Moreover, the different membrane morphological responses reveal that the two compounds interact with the phospholipids differently. The difference might be induced by the effects of lipid charge on the membrane translocation rates.

## 6. Applications: Gram-Negative Bacterial Membranes

### 6.1. Different Mimetic Gram-Negative Bacterial Outer Membranes

Phospholipid and LPS monolayers have been used to characterize the interaction behavior of antimicrobial compounds at the membrane interface and the extent of permeation into the membrane [117,118,119]. The monolayers on the Langmuir trough are compressed to 28 mN/m to represent a natural membrane pressure before assessing the interaction with antimicrobial compounds [120]. Pure PG and the mixture of PG/PE were utilized as a model Gram-negative bacterial membrane leaflet [61,120]. Recently, Gram-negative LPS outer membrane components such as Rc-LPS and Ra-LPS were successfully deposited at the air–liquid interface as stable monolayers for molecular-level mechanistic interactions [9,42,117]. Paracini et al. provided an overview of a Langmuir monolayer of LPS and supported model membranes containing LPS at interfaces as models for biophysical investigations of the Gram-negative bacterial outer membrane [6].

SLBs are simple and robust membrane mimics that are widely employed in the research of Gram-negative bacteria. The mixtures of PC with PG and PE have been commonly used as model membranes to elucidate the mechanism of antimicrobial agents [43,121,122]. PC lipids are not actually found in bacteria but are regularly used in bacterial membrane models due to their propensity to form good solid-supported bilayers [123]. Synthetic phospholipids are a popular method used to form mimetic Gram-negative bacterial membranes. For example, a study used mixtures of 1-palmitoyl-2-oleoyl-*sn*-glycero-3-phospho-(1′-rac-glycerol) (POPG) and 1-palmitoyl-2-oleoyl-*sn*-glycero-3-phosphatidylethanolamine (POPE) by vesicle fusion [18]. The amount of POPG was varied between 10 and 25 mol%, approximately close to the PG composition of Gram-negative membranes. By using the NR and QCM-D techniques, the study revealed that the optimal concentration of divalent ions, temperature, and continuous mass flow to the surface can achieve a full coverage of lipid bilayers on silica.

In addition to synthetic phospholipids, a recent study successfully formed SLBs composed of *E. coli* lipid extracts [69]. The study shows that bacterial lipids, which typically include LPS and non-polar lipids, can be extracted by applying a mixture of chloroform and methanol [124,125]. The total extract can be precipitated with acetone and extracted with diethyl ether to remove non-polar lipids [69]. The polar lipid extracts of *E. coli* contained approximately 75% PE, 13% PG, and 12% CL [69], forming high-coverage lipid bilayers via a vesical fusion approach. It is a more advanced membrane model as it contains natural CL, which is close to the complexity of Gram-negative bacterial membranes. However, LPS molecules are highly water soluble, meaning a large amount can be lost during the lipid extraction process [69]. Therefore, SLBs of *E.coli* lipid extracts are unable to represent the real compositions of bacterial membranes.

Clifton et al. showed the validity of incorporating lipid A, Rc-LPS, and Ra-LPS in the fabrication of complex asymmetric outer membrane models [8]. Asymmetric bilayers were formed using the LB/LS dipping approach. The inner leaflet was composed of DPPC with the capacity of forming stable bilayers on silicon wafers. To analyze the interfacial structures, a five-layer model was fitted to the model bilayers [8]. The five layers spanning from silicon to the bulk solution were the silicon oxide layer, inner leaflet headgroups, inner leaflet acyl chains, outer leaflet acyl chains, and outer leaflet headgroups [8]. The thickness of the outer leaflet headgroup showed the trend of lipid A < Rc-LPS < Ra-LPS. Furthermore, a Gram-negative bacterial outer membrane model using DPPC as the inner leaflet and Ra-LPS as the outer leaflet was assayed for the stabilizing effects of divalent cations, Ca^2+^; this research revealed that divalent cations were critical for membrane stabilization, resulting from the observation that Ca^2+^ can bind to the core region’s Ra-LPS films [9]. In addition, asymmetric DPPC/LPS (Rd or Ra) bilayers were fabricated to explore the effects of core oligosaccharide size on the electrostatic interactions between the antimicrobial proteins and the outer membrane [126]. The results revealed that the outer core region weakens the nonspecific electrostatic binding of the protein to the outer membrane surface [126].

Except for the solid support bilayers, floating supported bilayers have been advanced for improving the fluidity of the model membranes by preventing the bilayer from being in direct contact with the solid–liquid interface. The asymmetric membrane of DPPC and Ra-LPS was produced, which rested on a water layer above the ω-thiolipids and self-assembled monolayer coated gold surface [77,78]. Experimental results show that the asymmetric “floating” mimetic membrane that used the rough mutant Ra-LPS extracted from *E. coli* agrees well with the simulation results, which means that the approach can be used to explore the characterizations of the bacterial outer membrane [78]. Table 1 below summarizes some recent studies that have used model Gram-negative bacterial membranes and their applications. The table includes three model membrane types: monolayer, supported symmetric bilayer, and asymmetric bilayer, as well as the compositions, applications, and observations associated with the model membranes. Examples with different compositions are summarized here, intending to provide a brief overview of the current common compositions used in mimicking Gram-negative bacterial membranes. The observations cover the thickness, roughness, and coverage of the model membrane, surface pressure of the monolayer, asymmetry ratio of the asymmetric bilayer, and some significant findings.

### 6.2. Interaction of Antibiotics with Gram-Negative Bacterial Membranes

Various mimetic membranes are widely used to investigate how antibiotics interact with Gram-negative bacterial membranes and to understand why bacteria induces resistance. Polymyxins are considered to be the last-line antibiotics against the Gram-negative pathogen of *Pseudomonas aeruginosa*, which target lipid A on the bacterial membranes [128]. In recent research, it was discovered that polymyxin B can induce lipid A deacylation, resulting in polymyxin resistance [129]. To further understand the mechanism, Han et al. created two different mimetic Gram-negative bacterial membranes models of penta-acylated lipid A (resistance membrane) and hexa-acylated lipid A (wild-type membrane) for polymyxin B interactions. Neutron results suggested that penta-acylated lipid A reduced the polymyxin B penetration into cellular membrane, causing polymyxin B resistance [129]. In addition, octapeptin A3, a superior antimicrobial lipopeptide against polymyxin-resistant Gram-negative bacteria, was used to investigate the antimicrobial activity of octapeptin A3 with lipid A [128]. Octapeptin A3 was found to compromise Gram-negative bacterial outer membranes by interacting with lipid A on the surface of the mimetic membrane initially, followed by all of the octapeptin A3 penetrating the fatty acyl core [128]. Furthermore, a Gram-negative bacteria mimetic membrane containing DMPC and DMPG (4:1 molar ratio) was applied to uncover knowledge concerning the mechanisms of two well-known aminoglycosides, kanamycin A and neomycin B [121]. The research showed that kanamycin A bound to the mimetic membrane only until its concentration was beyond a threshold value, which then started to disrupt the membrane [121]. Neomycin B showed the ability to destroy the mimetic membrane at all of the concentrations studied, from 1 to 15 µM [121].

Antimicrobial lipids also exhibit potent antibacterial activity against Gram-negative bacteria, especially capric acid (C-10) and its monoglyceride derivative, monocaprin [130]. Aside from the study that exploited SLB composed of synthetic lipid compositions to investigate capric acid and monocaprin [131], Hyldgaard et al. explored the antimicrobial mechanism of monocaprylate using SLBs composed of *E.coli* lipid extracts [132]. Polar lipid extracts (67% PE, 23.2%, and 9.8% CL) were utilized instead of total lipid extracts because 17.6% of the total lipid extracts were not characterized. The changes in mass and viscoelastic properties of SLBs upon addition of monocaprylate were monitored by the QCM-D technique. There were no significant changes in the frequency shift and energy dissipation until the addition of monocaprylate of up to 5 mM. The data indicated that the mass of SLB increased two to four times with an increase in thickness, and the viscosity decreased. Therefore, the *E.coli* lipid bilayer becomes more hydrated and less rigid after monocaprylate treatment, resulting in membrane permeabilization.

### 6.3. Bacterial Outer Membrane Protein Complex

In addition to probing the interactions with antimicrobial compounds, model bacterial membranes are useful for studying the dynamics of membrane protein complexes. In the outer membrane of Gram-negative bacteria, most proteins have β-barrel structures [133,134]. The insertion and assembly of β-barrel proteins into the plane of the outer membrane is catalyzed by the β-barrel assembly machinery (BAM) complex and the translocation and assembly module (TAM) [135,136]. TAM is composed of two proteins: TamA and TamB. The movement and activity of the TAM was dissected by the reconstitution of TAM in model membrane of 1-palmitoyl-2-oleoyl-sn-glycero-3-phosphocholine (POPC). Neutron data revealed that the initiation of movements of TamA was induced by the presence of a substrate, Ag 43 [135]. Chen et al. investigated the relative positions of BAM subunits in a membrane environment and their conformational changes via the addition of a substrate, PET [137]. The results demonstrated that BAM subunits changed the conformation through the addition of PET and then inserted into the mimetic POPC membrane [137].

Besides the nanomachines used for protein assembly, there have also been other membrane proteins in biomimetic membranes that have been studied. For instance, the self-assembly behavior of the mechanosensitive channel of large conductance, MscL was elucidated in the model lipid systems [138]. MscL was purified and reconstituted into deuterated DMPC vesicles at a ratio of 1:60 (*w/w*). The DMPC/MscL vesicles were deposited to a DMPC bilayer on a silicon substrate and the spatial distribution of channels in lipid-protein bilayers was subsequently explored by a neutron reflectivity analysis. The findings demonstrated that MscL formed a cluster in the DMPC bilayer [138].

### 6.4. Interaction of Nanoparticles with Gram-Negative Bacterial Membranes

Lyotropic liquid crystal nanoparticles (known as cubosomes) are classified as a new generation of nonantibiotics. The nanostructures of cubosomes are complex, including a curved lipid bilayer and nonintersecting water channels [139]. Cubosomes are thermodynamically stable structures with high loading volumes of hydrophilic, hydrophobic, and amphiphilic compounds and have great potential in drug delivery, such as for antimicrobials [139,140,141]. The interactions of cubosomes with symmetric LPS-deficient outer membrane models (PE, PG, and CL at the ratio of 64:26:10) and asymmetric wild-type outer membrane models (DPPC:Ra-LPS) were investigated using an NR analysis [47]. Cubosomes interacted with LPS-deficient outer membranes, causing serious damage. On the other hand, cubosomes only appeared outside of Ra-LPS headgroups upon the addition of cubosomes to the asymmetric wild-type DPPC:Ra-LPS outer membrane. Therefore, cubosomes cannot effectively disrupt the outer membrane of Gram-negative bacteria as a monotherapy. Lai et al. found that the combination of polymyxin B and cubosomes enhanced the bactericidal activity against Gram-negative bacteria [48]. To elucidate the mechanism of the polytherapy, NR was used to investigate the interactions between cubosomes and polymyxin B with the supported asymmetric bilayer model featuring lipid A as the outer leaflet and 1,2-dipalmitoyl-d_62_-sn-glycero-3-phosphocholine (d-DPPC) as the inner leaflet. A negligible change in membrane structure was observed after adding 4 µg/mL polymyxin B to the cubosome-treated model membrane. In contrast, significant changes in membrane structure were observed with a significant loss of membrane material from the surface by the addition of cubosomes to the model membrane after a low concentration treatment of polymyxin B. The NR results unveiled the key interaction mechanisms demonstrating that polymyxin B destabilized the lipid at the outer leaflet and the bacterial outer membrane was subsequently disrupted by cubosomes [48]. The membrane mimetic system can clearly explain bacterial killing mechanisms in vitro.

## 7. Conclusions

This review summarizes some of the most commonly supported biomimetic bacterial membrane models with the corresponding synthetic methods. The realism of these membrane mimetics increases when proceeding from the monolayer to the solid-supported bilayer. The phospholipid monolayer is suitable for investigating interactions at bacterial membrane interface and the extent of the permeation of compounds. The lipid bilayer models can be characterized as symmetric bilayers and asymmetric bilayers, depending on the lipid compositions of the inner leaflet and outer leaflet. The asymmetric bilayers are typically used to emulate the Gram-negative bacterial outer membrane. Compared with the asymmetric bilayer model, solid-supported symmetric lipid bilayers are simple and robust to fabricate, with less technique requirements. To further improve the realism, floating supported membrane models are constructed to reduce the friction between lipid headgroups and solid substrates. However, it is not necessary to equate the model with the highest similarity with the best model for investigating the properties, structures, and transport mechanism across the membrane; various models have been reported to successfully mimic bacterial membranes using a range of lipids and ratios, and the model of choice is tailored to the problem that is being addressed. To achieve more realistic models, the lipid compositions are modified with real lipids (PG, PE, and CL) in Gram-negative bacterial outer membranes. Pure PG and mixtures of PC/PG/PE, PE/PG, and PE/PG/CL are utilized to mimic the Gram-negative bacterial inner membrane. Until recently, rough mutant LPS molecules (lipid A, Rc-LPS, Ra-LPS) and polar lipid extracts from *E. coli* have been used to represent the upper leaflet of Gram-negative bacterial outer membrane. However, *E. coli* membrane extracts lose LPS during model membrane fabrication, so the model composed of bacterial cell membrane extracts cannot effectively represent real compositions. The upper leaflet of rough mutant LPS molecules coupled with the inner leaflet of DPPC is the most common mimetic Gram-negative bacterial outer membrane to date. Gram-positive bacterial membranes are generally predominated by PG and CL. Pure PG and mixtures of PC/PG, PE/PG, and PG/CL at various molar ratios are therefore commonly used for model Gram-positive bacterial membranes. The mixture of PG/CL/L-PG at the molar ratios of 69:12:19 and 23:60:17 were the most accurate lipid compositions for the biomimetic Gram-positive membrane derived from the lipidomics data. Lipoteichoic acid is an important component of Gram-positive bacterial membranes. The membrane mimetics containing lipoteichoic acid have yet to be explored.

To understand why bacteria induces resistance, appropriate characterization techniques are employed to study the interactions of antimicrobials with the model membranes, such as the QCM-D, SPR, and NR techniques. These complementary techniques provide information on the change of membrane thickness, roughness, coverage, location of compounds, and binding affinity. In addition to the exploration of the interactions with antimicrobials, the dynamics of membrane proteins can be probed by using the reconstituted proteins in the model membranes. In recent years, significant progress has been made to understand and study the potency of alternative drug candidates to conventional antibiotics, including antimicrobial peptides, antimicrobial lipids, and lipid nanoparticles. Biophysical studies of novel antibiotics have been conducted to reveal their mechanisms and interactions with membranes. However, there are only a few studies on the mechanisms of antimicrobial lipids using supported lipid bilayers, and they have mainly focused on several medium-chain fatty acids and monoglycerides. It is also noticeable that the supported bacterial membrane mimetics in biophysical studies are unable to fully represent the bacterial membranes because of the simplified compositions and structures. The floating double bilayers eliminate the constraints exerted by the substrates and is most relevant to the Gram-negative outer membrane structure, but this model has not been widely applied in this area. Therefore, it is possible to conduct more biophysical research on new emerging antimicrobial agents using supported bacterial membrane mimetics and more realistic models should be further developed to convey the actual interactions between antimicrobials and bacterial membranes. Overall, establishing combinations of suitable bacterial membrane models and characterization methods are useful methods for designing new drugs that combat antimicrobial resistance.

## Figures and Tables

**Figure 1 membranes-12-00906-f001:**
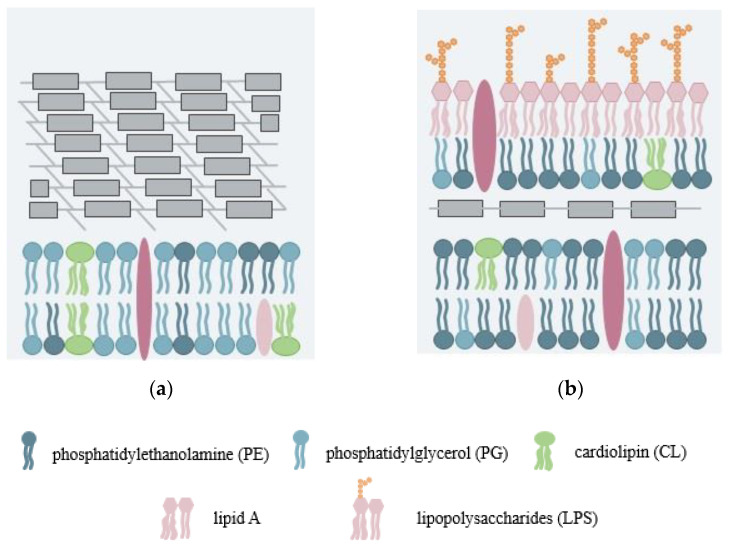
A schematic representation of (**a**) Gram-positive and (**b**) Gram-negative bacterial membranes.

**Figure 2 membranes-12-00906-f002:**
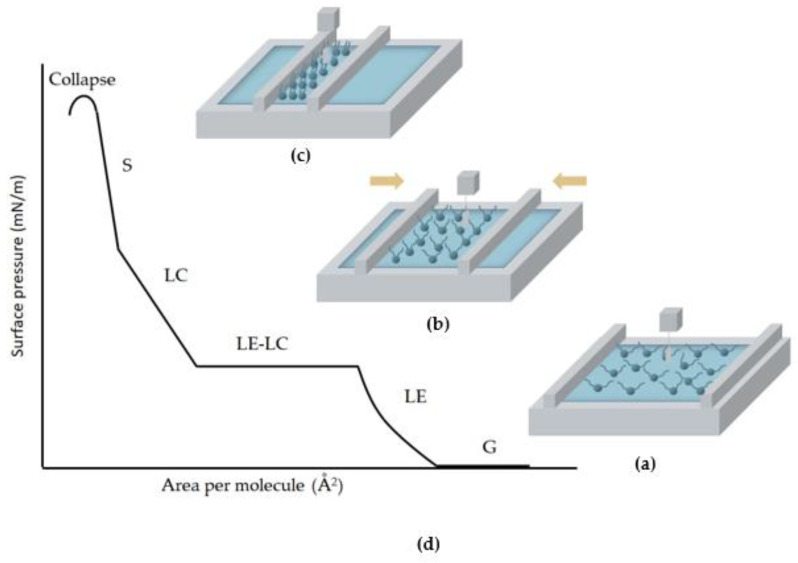
A schematic representation of the Langmuir monolayer formation process. (**a**) The lipid molecules are dissolved in an organic solvent and dispersed on the aqueous surface. The hydrophobic tails remain in the air with large tilt angles facing away from normal to the interface. (**b**) The two mobile barriers start to compress at a constant rate, and the Wilhelmy plate monitors the pressure. (**c**) Finally, the monolayer is formed with the molecules being perfectly organized. (**d**) A schematic of the π-A isotherm of the phospholipids. G: gaseous phase; LE: liquid expanded; LC: liquid condensed; and S: solid. During the compression process, the monolayer undergoes several phase transitions, from gaseous, to liquid, to solid states, and finally is perfectly self-oriented at the interface. Different phases are attributed to different degrees of freedom at various molecular organizations. Initially, the monolayer is in the gaseous phase with few interactions between the molecules. As the monolayer is compressed, the transition from the gaseous to liquid-expanded state occurs. The monolayer starts to be coherent, but the tail groups of the phospholipids are randomly oriented. As the area per molecule is further reduced, there are condensed lipid domains present in the expanded phase. Under further compression, the condensed phase is formed, where the monolayer exhibits a well-defined in-plane structure with a strong lateral cohesion. If the compression continues to be applied after the molecules are perfectly oriented, a collapse occurs due to mechanical instability [29].

**Figure 3 membranes-12-00906-f003:**
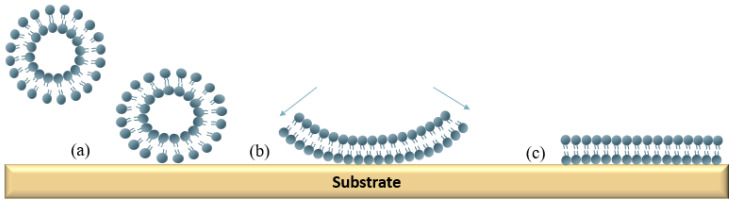
A schematic presentation of the vesicle fusion process for the formation of the supported lipid bilayer (SLB). (**a**) Small unilamellar vesicles are adsorbed to the substrate surface. (**b**) The fusion of small unilamellar vesicles occurs. (**c**) A continuous lipid bilayer is formed by the spread of the small patches.

**Figure 4 membranes-12-00906-f004:**
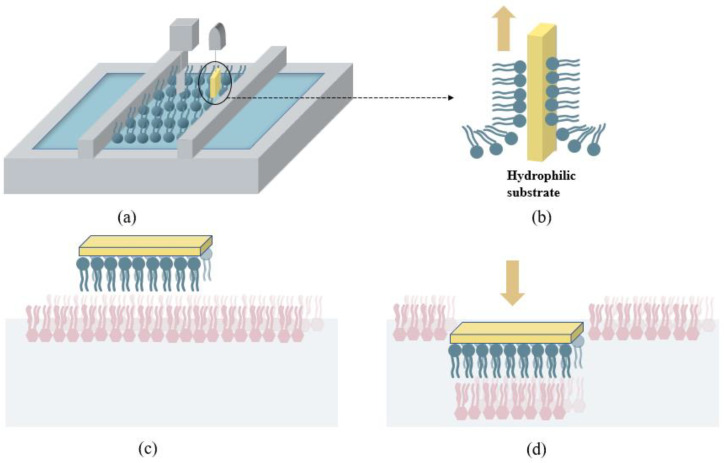
A schematic representation of vertical Langmuir–Blodgett and horizontal Langmuir–Schaefer depositions. (**a**) The phospholipid monolayer is compressed to the desired surface pressure at the air–water interface. (**b**) The phospholipid monolayer is transferred to the hydrophilic substrate by vertically lifting the substrate. (**c**) The Langmuir–Blodgett deposited monolayer is suspended above the air–liquid interface of the LPS monolayer. (**d**) The silicon crystal with the deposited Langmuir–Blodgett film is lowered to the interface to form bilayer with LPS monolayer.

**Figure 5 membranes-12-00906-f005:**
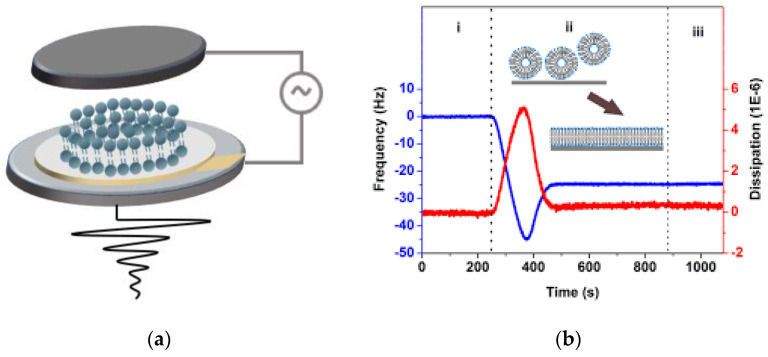
(**a**) Schematics of the quartz crystal microbalance method. The crystal resonates in shear mode. The shear wave in the sample and the environment, generated by the motion of the crystal, decays in a short distance due to energy loss via dissipative processes. (**b**) The representative frequency (blue) and dissipation shifts (red) of QCM−D during the formation of the SLB [85]. (i. Rinsing the sensor chip with buffer ii. Bilayer formation iii. Rinsing the bilayer with buffer). Reprinted with permission from [85], Copyright 2019 ACS Publications.

**Figure 6 membranes-12-00906-f006:**
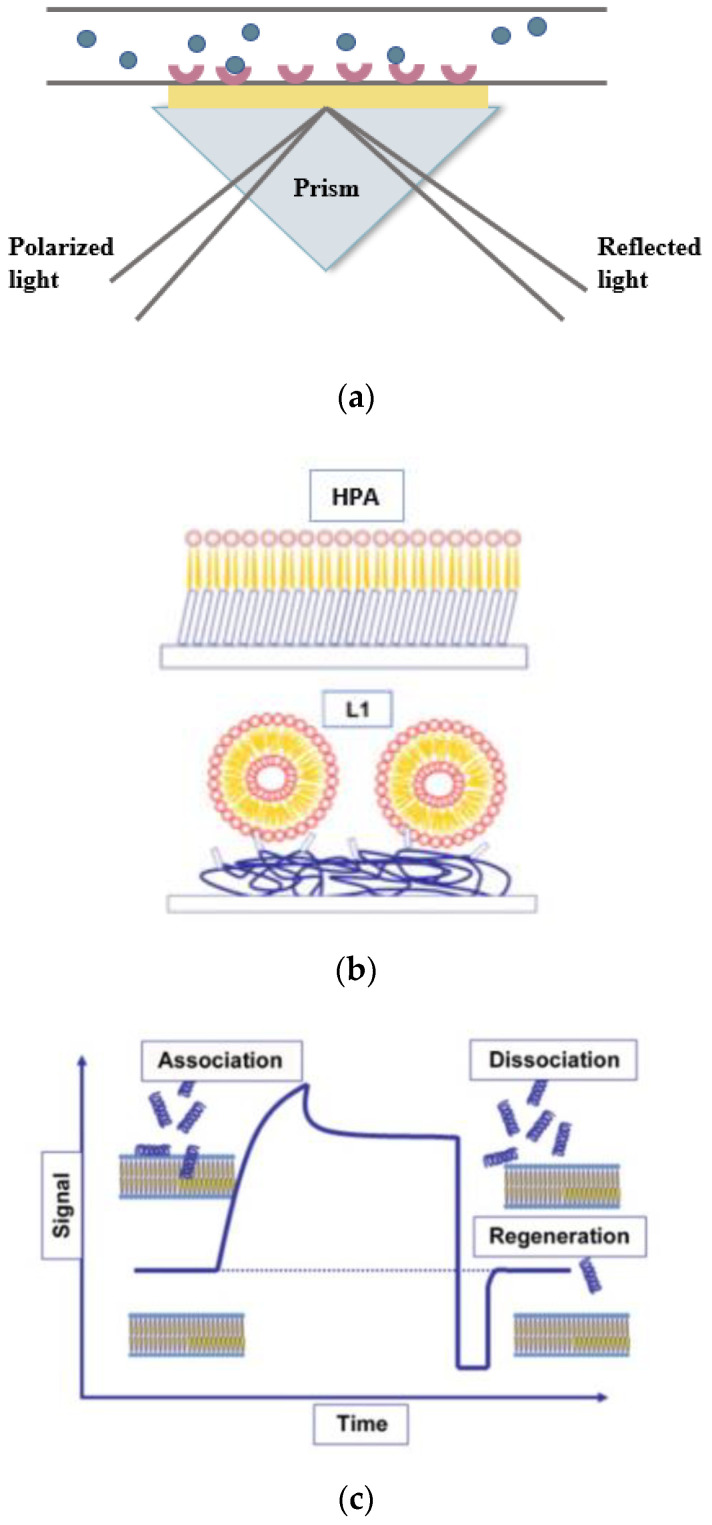
(**a**) The geometry of the surface plasmon resonance instrument (**b**) A schematic presentation of the HPA and L1 chips, adapted from [91]. (**c**) A sensorgram of the peptide–membrane binding including the association phase followed by the dissociation phase, reprinted with permissionfrom [91] under the Creative Commons Attribution 3.0 license (https://creativecommons.org/licenses/by/3.0/ (accessed on 30 May 2022)).

**Figure 7 membranes-12-00906-f007:**
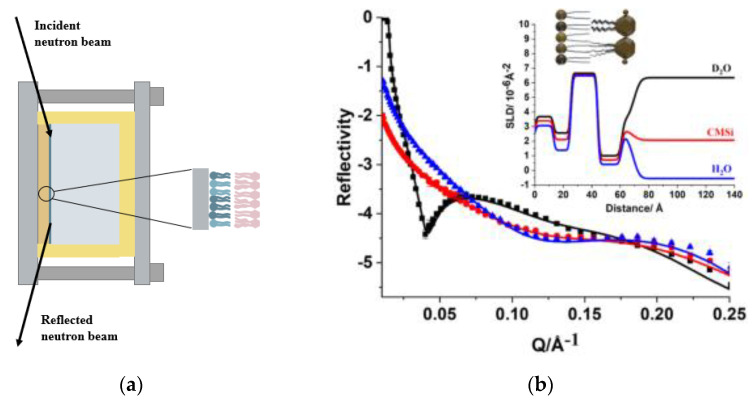
(**a**) A schematic representation of the NR method. (**b**) The NR profiles with the corresponding SLD graphs (top right corner) under D_2_O (black) CMSi (red) and H_2_O (blue) contrasts. The NR technique was conducted on a model Gram-negative bacterial outer membrane of d−DPPC (inner leaflet)/ lipid A (outer leaflet) on the SiO_2_ surface. The model membrane was characterized in three isotopic contrasts, i.e., in D_2_O, CMSi (contrast-matched silicon), and H_2_O buffers. Adapted with permission from [48] under the Creative Commons Attribution 4.0 license (https://creativecommons.org/licenses/by/4.0/ (accessed on 30 May 2022)).

**Table 1 membranes-12-00906-t001:** Summary table for the recent studies exploiting Gram-negative bacterial membrane models.

Model Membrane Types	Bacterial Membrane System	Applications	Observations	Ref
monolayer	DPPG	Interaction with antimicrobial peptides	Surface pressures increased from 28 mN/m to 43 mN/m after a subphase injection of lipopeptide. The thickness of tail and head layers were 18 and 16 Å, respectively. DPPC had a high coverage with an area per molecule of 48 Å^2^. The monolayer was thickened after lipopeptide injection. After the subphase injection of C_8_G_2_, approximately 15% of the DPPG molecules were desorbed and 10% formed a lipopeptide/lipid layer of 19 Å under the headgroup layer.	[120]
	Ra-LPS	Effects of divalent cations Ca^2+^	**In the presence of Ca^2+^:** Thickness of: Ra-LPS tail: 13.8 ± 0.1 Å Inner core oligosaccharide: 23.5 ± 0.5 Å Outer core oligosaccharide: 8.6 ± 0.5 Å **In the presence of EDTA:** Thickness of: Ra-LPS tails: 12.7 ± 0.2 Å Inner core oligosaccharide: 21.2 ± 0.5 Å Outer core oligosaccharide: 8.5 ± 0.2 Å There were 5.3 Ca^2+^ per Ra-LPS headgroup. The presence of EDTA rendered a less ordered Ra-LPS monolayer.	[9]
	Ra-LPS/DPPC	Interaction with antimicrobial peptides (LL37, LFb)	The initial surface pressure was 25 ± 4 mN/m and after the injection of LL37 and LFb, the P_max_ was 18.42 mN/m and 17.42 mN/m, respectively. Limited penetration of peptides in the tail region of the monolayer. Thickness of the tai was 14.0 ± 0.8 Å. Thickness of the inner headgroup layer was 15.0 ± 0.2 Å. **After adding peptides**: Thickness of LL37 peptide layer was 261.7 ± 50.2 Å and the thickness of LFb was 30.5 ± 2.7 Å. 40.6% of LL37 and 23.7% of LFb within the outer headgroup region. The Outer headgroup layer thickness increased to 20.0 ± 0.5 Å 14% of LL37 within the inner headgroup	[117]
	Rc-LPS/DPPC	Interaction with antimicrobial peptides (LL37, LFb)	The initial pressure was 28 ± 2 mN/m and after injection of LL37 and LFb, the P_max_ was 11.24 mN/m and 15.56 mN/m, respectively. The area per molecule of the Rc-LPS was 119.7 ± 7.3 Å^2^. Thickness of the tail region before adding peptides was 14.0 ± 0.8 Å. There was a limited penetration of peptides in the tail region of the monolayer Inner head group layer info: Thickness: 9.9 Å; Roughness: 4.4 ± 0.5 Å; Coverage: 59.6 ± 6.5% **After** **adding LL37 or LFb peptides**: 41.7% of LFb was found in the inner headgroup region. Thickness of outer headgroup layer was 20.1 ± 1.3 Å 22.8% of LL37 and 8.1% of LFb were found in the outer headgroup. Thickness of LL37 was 323.4 ± 60.1 Å and thickness of LFb was 98.7 ± 21.3 Å.	[117]
	Rc-LPS	Interaction with antimicrobial peptides (G_3_, C_8_G_3_)	Thickness of tail layer was 13 ± 1 Å and head layer was 23 ± 2 Å. G_3_ displaced 34% of LPS molecules and 12% of LPS molecules aggregated with G_3_ formed a suspended stack layer outside the headgroups (thickness: 36 ± 3 Å). C_8_G_3_ removed 51% of LPS molecules and 4% LPS aggregated with C_8_G_3_ formed suspended stack layer outside the headgroups (thickness: 25 ± 2 Å).	[42]
Symmetry bilayer	POPE/POPG	NA	This was the first time to form POPG/POPE supported lipid bilayers by using vesicle fusion approach. The optimal conditions for bilayer formation was 50 °C temperature, continuous flow, pump rate of 1 mL/min, and 0.1 mg/mL lipid vesicles in 3 mM CaCl_2_. At 50 and 37 °C, the bilayer of 75 mol% POPE and 25 mol% POPG achieved full coverage. The sizes of headgroups and tail were 5.5 ± 0.5 Å and 31 ± 0.5 Å, respectively. The tail core thickened to 1 Å when the temperature decreased from 37 to 25 °C	[18]
	Lipid extracts of *E. coli*	NA	The compositions obtained for total *E. coli* lipid extracts from hydrogenated and deuterated *E. coli* bacteria were similar (approx. 75% PE, 13% PG, and 12% CL) The thickness of high-coverage bilayer was 41 ± 3 Å (hydrogenated E. coli headgroup: 7 ± 1 Å, tail region: 27 ± 1 Å)	[69]
	DMPC/DMPG	Interaction with LL-37-loaded cubosomes	The bilayer achieved high coverage of 4.3 ± 0.1 mg/m^2^ with a mean molecular area of 57.0 ± 1 Å^2^. The total thickness was 43.7 ± 1 Å. (headgroups: 8.0 ± 0 Å; tail region: 27.7 ± 1.0 Å) After the introduction of LL37-loaded cubosomes, the thickness increased to 49.4 ± 2 Å.	[43]
	POPE/POPG/TOCL	Interaction with antimicrobial peptides (colistin)	The tethered bilayer thickness was approx. 19 Å. The area/unit cell (A_L_) decreased at 500:1 lipid/colistin and increased at 200:1 lipid/colistin. The bending modulus (Kc) of mimetic inner membrane: colistin (100:1 molar ratio) was 7.5 $\times 10^{-20}$ J. Colistin reached the interior of the hydrocarbon region in Gram-negative inner membrane mimics.	[127]
	LPS/DLPG	Interaction with antimicrobial peptides (colistin)	The tethered bilayer thickness was approx. 20 Å. The area/unit cell remained fairly constant as the ratio of lipid/colistin changed. The bending modulus of mimetic outer membrane: colistin (100:1 molar ratio) was 2.0 $\times 10^{-20}$ J. Colistin located in the headgroup region in Gram-negative outer membrane mimics.	[127]
Asymmetry bilayer	Ra-LPS	NA	The floating supported bilayer was approximately 15 Å above the head group of choline. d-DPPC head-group layer: 14–18 Å and outer head-group region: 28–30 Å. The asymmetry (LPS/PC) of the outer leaflet ranged from 69:28 to 79:11 and that of the inner leaflet ranged from 23:75 to 8:82. The coverages of bilayers were above 90%. After removing Ca^2+^ by EDTA, the bilayer asymmetry was lowered by approximately 20%. After the introduction of lactoferrin (40 μg/mL) to the outer membrane model, the coverage was reduced by 12% and the asymmetry was reduced by approximately 30%.	[77]
		Effect of divalent cations	**In the presence of Ca^2+^ solution**: The asymmetry (LPS/ DPPC) of the outer leaflet ratio: 63:32 and the inner leaflet ratio: 37:59. Thickness of inner headgroup (DPPC) was 13.0 ± 0.9 Å, inner tail was 17.0 ± 0.2 Å, outer tail was 14.5 ± 0.7 Å, and outer headgroup was 31.0 ± 1.0 Å. Bilayer coverage: 96 ± 4% **In the presence of EDTA solution**: Thickness of inner headgroup (DPPC) was 15.4 ± 4.0 Å, inner tail was 15.9 ± 1.0 Å, outer tail was 11.0 ± 5.0 Å, and outer headgroup was 28.4 ± 1.0 Å. Coverage: 96 ± 4%	[9]
		NA	The asymmetry (Ra-LPS/ DPPC) of the outer leaflet ratio: 67:22 and inner leaflet ratio: 19:66 Thickness of inner headgroup was 14.8 ± 2.0 Å, inner tails was 15.6 ± 0.6 Å, outer tails was 16.0 ± 4.8 Å, and outer headgroup was 31.0 ± 1.2 Å. d-DPPC: Ra-LPS bilayer coverage: approx. 85%	[8]
	Rc-LPS	NA	The asymmetry (Rc-LPS/ DPPC) of the outer leaflet ratio: 25:58 and the inner leaflet ratio: 28:57 Thickness of inner headgroup was 8.4 ± 11.2 Å, inner tails was 18.2 ± 2.5 Å, outer tail was 15.3 ± 3.0 Å, and outer headgroup was 20.9 ± 2.0 Å. d-DPPC: Rc-LPS bilayer coverage: 84 ± 5%	[8]
		Interaction with antimicrobial peptides (G_3_, C_8_G_3_)	Thickness of inner head layer was 8 ± 1 Å, inner tail layer was 17 ± 1 Å, outer tail layer was 14 ± 1 Å and outer head layer was 26 ± 2 Å. The area per molecule of inner leaflet (DPPC): 58 ± 6 Å^2^ and outer leaflet (LPS): 162 ± 16 Å^2^. Upon the injection of G_3_, 17 ± 2% of DPPC and 37 ± 3% of LPS molecules were removed and 13 ± 1% DPPC and 10 ± 1% LPS molecules became a suspended stack layer, aggregated with G_3_ with thickness of 88 Å. Upon the injection of C_8_G_3_, 61% of LPS molecules were removed and 38% of LPS molecules became suspended stack layer, aggregated with C_8_G_3_ with thickness of 48 Å.	[42]
	Lipid A	NA	The asymmetry (lipid A: DPPC) of the outer leaflet ratio: 65:26 and the inner leaflet ratio: 36:55. Thickness of inner headgroup was 8.5 ± 1 Å, inner tail was 19.8 ± 2 Å, outer tail was 17.6 ± 3.4 Å, and outer headgroup was 8 ± 5 Å. d-DPPC: lipid A bilayer coverage: 91 ± 5%	[8]

## Data Availability

The datasets used and/or analysed during the current study are available from the corresponding author on reasonable request.

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
