# Peer review of "Solid and Liquid Surface-Supported Bacterial Membrane Mimetics as a Platform for the Functional and Structural Studies of Antimicrobials"

_membranes, 2022, doi:10.3390/membranes12100906_

Round 1

Reviewer 1 Report (New Reviewer)

This review provides introduction to a few model membrane systems applicable is a few biophysical techniques (QCM-D, SPR, neutorn reflectometry) with very brief introduction to each techniques. The most useful part of the review is the summary tables of recent research using those techniques on membrane interaction with different antibiotics based on different model membrane system.

I think the English and the writing style can be improve to make the whole article more readable. Also since the introduction to the techniques are very brief in nature, it will be useful to explicitly to point to a few additional specific reviews on different techniques at each section. This will help readers to quickly identify a good source if they need to pursue further.

Author Response

Reviewer 2 Report (New Reviewer)

This is a very interesting review summarizing the development of bacterial-membrane-mimicking platforms to study antimicrobials. The manuscript is very well structured covering three types of model membrane platforms -- monolayers, supported lipid bilayers, and supported asymmetric bilayers -- and emphasizes various kinds of surface-sensitive measurement techniques such as QCM-D, SPR, and NR. I recommend that the manuscript is publishable pending minor revision to clarify some key concepts and to better cover some of the most promising antimicrobials that fit within the journal scope as follows:

  1. The major focus is synthetic lipid systems but what about the use of bacterial cell membrane extracts? Please add more detail about the use of such extracts, including compositions and how they compared to model compositions, and specific design challenges from a fabrication perspective.    2. The pros and cons of the different measurement techniques should be better described (compare/contrast) in terms of analytical merits.   3. The concept of antimicrobials should be better described and clarified in terms of the operating principles and scope covered, which seems to include peptides, antibiotics, and lipid nanoparticles such as cubosomes. In addition to clarifying this scope so it is well understood by the future readership, the scope should be expanded to include relevant studies involving medium-chain fatty acids and monoglycerides, which are quickly emerging as one of the most promising antimicrobial solutions.   4. Please comment on the potential of developing double-membrane mimetic structures that might be useful to test antimicrobials that could inhibit Gram-negative bacteria, for example. This currently seems to be a major gap in the field overall.   5. The conclusion should be improved to have a stronger, more critical outlook to describe potential routes of and needs for future research.

Author Response

Reviewer 3 Report (New Reviewer)

This paper presents well-known information on selected biomimetic membranes and three methods used for their characterisation.

1) The general structure of E. coli lipopolysaccharide is redundant (Fig. 1a).

2) lines 108-110.

"In this review, we only focus on lipid monolayers and solid-supported lipid bilayers because of their excellent electrical sealing properties, stability in air, and ease of formation and manipulating their lipid compositions". 

I disagree with the statement that lipid monolayers have excellent electrical sealing properties. In the case of bilayers, this is correct.

3) 116 line

The authors wrote "a variety of investigation techniques" and then listed only three techniques (the ones they later characterise in the manuscript)

4) Fig 2,3,4 - should be removed. They are elements of other drawings and, when presented in this form, may mislead the reader who is not familiar with monolayers or bilayers.

Fig 2, for example, shows a monolayer formed in the centre of the subphase surface. But what happens at the edges? Without a description of how the bath works, how the monolayer is formed, or a description of the barrier, etc., such a drawing is very misleading.

5) 162 line - "are simpler" than what?

6) 190 line - It does not necessarily have to be a PC.

7) subsection 3.1 - A drawing of the pressure-surface isotherm would be useful and its exact description (not neglecting the collapse).

8) Table 1 is too long.

9) A list of abbreviations would be useful.

1) The work is written carelessly, with lots of double spaces and red or underlined letters (e.q. 109, 447, 616, 701).

Author Response

Reviewer 4 Report (New Reviewer)

This manuscript reviews research activities related to the experimental investigation of bacterial membrane mimetics at planar solid or fluid interfaces with various surface-sensitive techniques. The manuscript provides a nice overview of the architecture and composition of natural bacterial membranes and of the composition of suitable simplified mimics, it describes the main methods to prepare such mimics, and it provides introductions into the most powerful surface-sensitive techniques, such as QCMD, SPR, and neutron reflectometry. Finally, it reviews studies that have used these techniques to investigate bacterial membrane mimetics and their interaction with antimicrobials. Overall, it is a helpful and practical introduction for newcomers to the research field, but also provides a good overview of the literature for more experienced researchers in the field. The manuscript is generally well written and comprehensible.

There are however a number of shortcomings that should be improved before publication.

1.) Reference 6 (Paracini et al.) is a recent review article. It would be a good service to the scientific community, if the authors would mention somewhere that this is actually a highly related review article, although its focus is slightly different (My impression is that it is very similar but with a focus solely on lipopolysaccharides, while the present manuscript addresses bacterial membrane mimetics in a broader sense).

2.) An important component of gram-positive bacterial membranes is lipoteichoic acid. But the manuscript does not mention any membrane mimetic studies dealing with this component. It would be good to make sure that the review is complete in this regard as much as possible.

3.) Technical details on neutron reflectometry: The authors call q the scattering vector but then give a scalar (i.e., non-vectorial) definition of it. What the authors probably mean is the perpendicular component, qz, of the scattering vector (which is the usual terminology in the community). Or alternatively mean with the scalar quantity q the magnitude of the scattering vector, which under specular conditions coincides with qz.

4.) Another technical subtlety: I do not agree that SLD is analogous to a refractive index. To my understanding the SLD determines (but is not proportional or even identical) to the refractive index for neutrons (n = 1 – c*SLD). But one could say the SLD modulates the refractive index of a material for neutrons.

Round 2

Reviewer 3 Report (New Reviewer)

The authors revised the manuscript according to the recommended suggestions. I now support the publication of the article.

This manuscript is a resubmission of an earlier submission. The following is a list of the peer review reports and author responses from that submission.

Round 1

Reviewer 1 Report

The topic is nice. However, in a review like this the authors need to cover many background information that are available to date. Too few references are quoted here. There are vital references and information thereof in the area missing. The article needs to get enhanced adding more information on especially model studies of various kinds. Otherwise, it will be injustice to writing a review article. The readers have to get a rigorous background in the field covered in any review article. 

Reviewer 2 Report

The authors aimed to describe three standard characterisation techniques to probe the properties of model bacterial membranes and their interactions with antimicrobials.

The review is well structured, and the topic is particularly interesting to the reader.

-      Despite this, it is necessary to significantly extend some paragraphs and cover the most recent literature.
In this regard, some papers have been suggested to expand the described sections.

https://link.springer.com/protocol/10.1007/978-1-0716-1166-1_5

10.1021/acs.langmuir.0c02516

DOI: 10.1021/acs.jafc.1c06747

DOI: 10.1016/j.apsb.2021.02.013

DOI: 10.1002/marc.202100194

-      Figure 1 must be redone. Authors should use a different method to save their images from improving the quality of the figures. Please keep the image without the red underline.

-      Please rewrite the sentence in lines 208-210 because the procedure is unclear.

-      Change the point (a) of the caption in figure 5 in this way:

The lipid molecules are dissolved in an organic solvent and dispersed on the aqueous surface.

-      Please check the grammar, formatting, style, and font size change (see table 1 or line 329).

-      Figure 9 needs to be redone; parts (b) and (c) are not adapted from reference 33 but are the same as figure 2 in paper 33.

-      Table 1 is very interesting but, unfortunately, very basic and poorly described. Please extend this part.

Reviewer 3 Report

The review summarizes the most common biomimetic bacterial membrane models with corresponding synthetic methods. I believe that this review is interesting and should be publishable in this journal; however there are several scientific aspects of this review that I feel the authors must first address.

1. The introduction section does not provide complete information about the choice of the presented model: lipid monolayer and supported lipid bilayers. Why do not used the consider other models, spherical liposomes and planar lipid membranes according to the Montal and Muller method. I recommend should be added to the review.

2. Please, clarify the choice of monolayers if the authors write about the impossibility of studying the effect of the antimicrobial peptides "the Langmuir monolayer is limited in the study of intermolecular behaviour, the effects and interactions of compounds such as antimicrobial peptides on the membranes, and also the impact of some parameters such as cations strength, or temperature".

3. The section 3 included a description of the membrane models. Please, clarify the choice of lipids composition using the various methods. The different lipids (DPPC, LPS, DMPC/DMPG) mimicking the same membrane bacteria's?

4. The section 5 is very poor and must be rewriting and include the more details information and major aim of the review (not only daptomycin).

5. Authors declare the study of the interaction of the antimicrobial drugs with model membranes, while the section 5 is extremely depleted in this information. I recommend rewriting this section and addition more antibiotics, their mechanisms association with the membrane of Gram-positive membranes.

6. There is no explanation for table 1. Please, explain why the discussion of the table 1 is not presented. What conclusions should the reader make from the table 1?

7. I recommend rewriting the abstract to include the more summary information and major conclusions of the review work.

8. Moreover, my great concern is related to the conclusions. They are too short and schematic. They should be modified. The most important findings of this work should be supported by results and their biological significance should be clearly specified. Moreover, I recommend add the summary figure or schema.

Round 2

Reviewer 1 Report

The article got improved, but not to my satisfaction. You have covered too many things without detailing various aspects as required in a review article. Moreover, The article is more like a chapter of a book based on mostly established facts. A review article requires more of the results-based discussion on topic's various aspects. Very little amount of results from published articles have been incorporated.

The conclusion is too big. 

Still more references are needed.  

The authors need to read various other full review articles before revising their's so that they know how to write a good review article.

Grammatical errors have been found quite a lot.

Author Response

We thank for reviewer for the constructive suggestions. The manuscript’s scope could be appropriately fit into this special issue’s requirements – “To cover novel research trends in the development of new membrane models or the use of membrane models for biological applications”.
Specifically, the scope of our review is to explore the use of liquid- and solid- surface supported lipid layer as mimetic bacterial membranes and their applications in exploring antimicrobial activities, which we feel fits within the boundaries of the special issue.  We have also added more results throughout the manuscripts and the refences numbers have increased from 60 to 124. The Grammatical errors were corrected.

Reviewer 2 Report

The authors have made the requested changes.

Author Response

Thank you

Reviewer 3 Report

The review summarizes the most common biomimetic bacterial membrane models with corresponding synthetic methods. Despite on the questions and remarks earlier, the authors neglected many of them and do not improved the text. I think that the manuscript in the Review category should be improved and reviewed.

The most significant key points:

1. The new title ("Supported Bacterial Membrane Mimetic in Functional and Structural Studies of Antimicrobials") is not reliable and can be revised.

2. Paradoxically the authors did not address this Review the different information about model membranes mimicking the bacterial membrane and improved only one world in the sentence: " In this review, we only focus on lipid monolayer and supported lipid bilayers. " without explanation. I am not convinced that the for the author's claims for a Review of an interesting area of the model lipid membrane mimicking the different membrane, this is a right decision.

3. Why are there no references in this sentence "Biomimetic bacterial membranes include lipid monolayer, supported lipid bilayers, spherical liposomes, and free-standing planar lipid membranes prepared by Montal and Mueller method"?

4. The reference list included a review of the literature data over the last 10 years (about 70% of reference list including paper for the last two-three years). Why authors did not use the original method paper and do not addition this information? This is not good reviews in this area. I recommend rewriting and addition of the information for the improvement the paper in the categories Review.

Author Response

 1.The new title ("Supported Bacterial Membrane Mimetic in Functional and Structural Studies of Antimicrobials") is not reliable and can be revised.

Thank you for the reviewer’s suggestions. We changed the title to be very specific - “ Solid- and Liquid-Surface Supported Bacterial Membrane Mimetics as a Platform for Functional and Structural  Studies of Antimicrobials”

  1. Paradoxically the authors did not address this Review the different information about model membranes mimicking the bacterial membrane and improved only one world in the sentence: " In this review, we onlyfocus on lipid monolayer and supported lipid bilayers. " without explanation. I am not convinced that the for the author's claims for a Review of an interesting area of the model lipid membrane mimicking the different membrane, this is a right decision.

Please refer to comment 1 above and the explanations of our focus was described in line 118-120, page 4. Specifically, with have changed the tile and outlined that the review focuses on liquid- and solid- supported surface bacterial membrane mimetics. 

  1. Why are there no references in this sentence "Biomimetic bacterial membranes include lipid monolayer, supported lipid bilayers, spherical liposomes, and free-standing planar lipid membranes prepared by Montal and Mueller method"?

The references for lipid monolayer (ref 12-14), supported lipid bilayers (ref 15-18), liposomes (ref 19-21), and Montal and Mueller (ref 22-24) were added

  1. The reference list included a review of the literature data over the last 10 years (about 70% of reference list including paper for the last two-three years). Why authors did not use the original method paper and do not addition this information? This is not good reviews in this area. I recommend rewriting and addition of the information for the improvement the paper in the categories Review.

The original references for some of the methods have been added where appropriate. For example, Montal and Mueller method in ref 22, solid-supported lipid bilayer in ref 69, LB-LS method in ref 50.

Round 3

Reviewer 1 Report

The manuscript is still as incomplete as it was in its first submission. My previous comments are still active. It lacks many components needed to qualify as a complete review article. 

Reviewer 3 Report

The authors have changed the title of the Review, but there is no significant improvement in the presented information about model membrane. It is seen the correction of the some sentences and slightly changed references to original paper. There are no significant changes of the Review. I think that the article "Review" mustld be improved according to all previous comments.